# Co-Administration of Aluminium Hydroxide Nanoparticles and Protective Antigen Domain 4 Encapsulated Non-Ionic Surfactant Vesicles Show Enhanced Immune Response and Superior Protection against Anthrax

**DOI:** 10.3390/vaccines8040571

**Published:** 2020-10-01

**Authors:** Himanshu Gogoi, Rajesh Mani, Anshu Malik, Parveen Sehrawat, Rakesh Bhatnagar

**Affiliations:** 1Laboratory of Molecular Biology and Genetic Engineering, School of Biotechnology, Jawaharlal Nehru University, New Delhi 110067, India; himanshuhrc2010@gmail.com (H.G.); rajeshbms2@gmail.com (R.M.); anshumalik7@gmail.com (A.M.); sehrawat.parveen01@gmail.com (P.S.); 2VC Office, Banaras Hindu University, Varanasi 221005, India

**Keywords:** *Bacillus anthracis*, protective antigen domain 4, aluminium hydroxide nanoparticles, non-ionic surfactant vesicles, nanoformulation

## Abstract

Aluminium salts have been the adjuvant of choice in more than 100 licensed vaccines. Here, we have studied the synergistic effect of aluminium hydroxide nanoparticles (AH np) and non-ionic surfactant-based vesicles (NISV) in modulating the immune response against protective antigen domain 4 (D4) of *Bacillus anthracis*. NISV was prepared from Span 60 and cholesterol, while AH np was prepared from aluminium chloride and sodium hydroxide. AH np was co-administered with NISV encapsulating D4 (NISV-D4) to formulate AHnp/NISV-D4. The antigen-specific immune response of AHnp/NISV-D4 was compared with that of commercial alhydrogel (alhy) co-administered with NISV-D4 (alhydrogel/NISV-D4), NISV-D4, AHnp/D4, and alhydrogel/D4. Co-administration of NISV-D4 with AH np greatly improved the D4-specific antibody titer as compared to the control groups. Based on IgG isotyping and ex vivo cytokine analysis, AHnp/NISV-D4 generated a balanced Th1/Th2 response. Furthermore, AH np/NISV-D4 showed superior protection against anthrax spore challenge in comparison to other groups. Thus, we demonstrate the possibility of developing a novel combinatorial nanoformulation capable of augmenting both humoral and cellular response, paving the way for adjuvant research.

## 1. Introduction

Since Edward Jenner’s discovery of the first vaccine against smallpox, vaccines have brought a revolution in the medical world. Their use has resulted in a huge economic and social impact, by lowering disease burden and increasing life expectancy in a cost-effective manner. The central rationale behind creating a vaccine is to induce an immune response resembling infection caused by a pathogen while avoiding the undesired side effects. To elicit a desired immune response, a vaccine often requires the inclusion of an adjuvant, which enhances the innate immune mechanism. Often, the adjuvant assists with directing the specificity of an immune response [1,2]. Aluminium salts have been the choice of an adjuvant since 1926 after Glenny demonstrated the potential of alum-precipitated diphtheria toxoid to enhance the antibody production [3]. The family of aluminium adjuvant comprises of aluminium hydroxide, aluminium phosphate, and potassium aluminium sulphate/alum. Aluminium salts, due to their low cost and safety profile, have been used in more than 100 licensed vaccines around the globe [4]. Although the physical attributes of aluminium salts have been well characterised by Hem and White [5], the cellular and molecular mechanism through which these adjuvants influence the immune system remains elusive. Based on different *in vitro* and *in vivo* studies, aluminium salts have been hypothesised to act by either (i) the formation of a depot site and slow release of the antigen, (ii) chemokine and cytokine-mediated antigen presenting cell (APC) recruitment at the injection site, thereby enhancing antigen uptake and presentation, or (iii) the activation of caspase 1 and secretion of mature IL-1β and IL-18 in an NLRP3 (NOD-, LRR- and pyrin domain-containing protein 3)-dependent mechanism [6]. Aluminium salts have been associated with only a modest immune response in humans and mainly induce a Th2 immune response with no/negligible CD8^+^ T-cell response. In order to elicit the desired stimulation, aluminium salts have been used in combination with different adjuvants. For example, GSK’s AS04 adjuvant system comprises monophosphoryl lipid A (MPLA) adsorbed onto aluminium hydroxide and cervarix, which comprises MPLA and virus-like particles from human papilloma virus adsorbed onto aluminium hydroxide [7]. However, these vaccines have often been reported to be associated with unpleasant but generally tolerable side effects such as the occurrence of local pain, swelling, irritation at the site of injection due to the presence of microparticulate alhydrogel, and MPLA, a bacterial endotoxin [8,9]. Recent research has reported that aluminium hydroxide nanoparticles were able to generate a strong antigen-specific antibody response compared to its counterpart of micron range. Moreover, these particles also induced the secretion of antigen-specific T-cell immune response, making them a suitable adjuvant for intracellular pathogens as well [10,11].

With a progress in nanoscience, nanotechnology has benefited tremendously in the field of drug delivery and vaccine delivery [12]. Nanoscale materials such as liposome, emulsions, virus-like particles, ISCOMs (immune stimulating complexes), and polymeric particles have received attention as potential delivery vehicles as well as immunomodulators [13]. Nanocarriers prepared from phospholipids/liposomes have been extensively used for delivering drugs as well as antigens. Owing to their tuneable properties, liposomes with surface, fluidity, and size modifications have been prepared to promote specific and selective immune response. However, being composed of phospholipids, liposomes face the problem of shelf life stability as they tend to undergo oxidative degradation [14,15]. This issue was addressed by Baillie et al. when they demonstrated the use of non-ionic surfactant to prepare liposome-like vesicles [16]. These non-ionic surfactant vesicles (NISV) are self-assembling lamellar structures prepared from non-ionic amphiphiles with/without cholesterol as an additive. These vesicles bear the same characteristics as the liposomes: biodegradable, non-immunogenic, and capable of encapsulating biologically active cargo. However, their chemical stability and low cost of preparation make them a valuable and interesting adjuvant candidate for industrial manufacturers.

Combinatorial adjuvant formulation to enhance the adjuvant potential of alhydrogel has also been studied recently. Glaxo Smith Klein’s AS-04 is such an example. More such work involving combining immunostimulatory adjuvants such as TLR agonists along with alhydrogel to enhance the Th1 and Th17 response are under clinical trials [17].

*Bacillus anthracis* (BA), the causative agent of anthrax, is a spore-forming, Gram-positive, rod-shaped and facultative anaerobic bacterium. Although primarily it is a disease of the ruminants [18], humans have faced the wrath of anthrax since time immemorial [18]. Its notoriety as a potent bioterror agent was highlighted during the 2001 USA mail attacks where letters laced with anthrax spores were mailed to US congresspersons and media, leading to five deaths. The toxicity of BA is associated to a tri-partite exotoxin and an anti-phagocytic poly-γ-D-glutamic acid capsule [19]. The anti-phagocytic capsule helps the bacteria evade phagocytosis [20,21]. The exotoxin comprising of protective antigen (PA), lethal factor (LF), and edema factor (EF) are individually non-toxic, but PA in combination with LF leads to host cell death, while the combination of PA and EF leads to homeostasis imbalance [22,23,24]. After host invasion, the bacterial spores germinate into functional vegetative cells, leading to the secretion of the exotoxins PA, LF, and EF. PA, an 83 kDa protein binds to the host macrophage cell receptors TEM-8 and/or CMG2 [25,26,27], resulting in the clipping off of a 20 kDa fragment by furin-like proteases. Domain 4 of PA is responsible for binding with the host cell receptors and has been reported to be slightly immunomodulatory [28]. The monomeric 63 kDa fragments combine to form a heptameric/octameric pre-pore complex that facilitates the binding of LF and/or EF competitively to be translocated into the cytosol [29]. LF is a 90 kDa zinc metalloprotease enzyme and clips the N-terminus of mitogen-activated protein kinases, ultimately leading to macrophage death [30,31,32]. LF inactivates cytoplasmic MEK1 and MEK2 present in host cells, and it also inactivates MKK3, which phosphorylates p38 MAP kinase in macrophages and decreases the production of reactive oxygen species (ROS) and TNF-α (tumor necrosis factor) [33,34]. EF is an 89 kDa calcium and calmodulin- dependent adenyl cyclase that results in an elevated cAMP, resulting in homeostasis and massive edema [35,36]. The prophylactic approach against the disease consists of vaccination involving PA as the major immunogen [37]. The currently licensed anthrax vaccines, anthrax vaccine adsorbed (AVA) and anthrax vaccine precipitate (AVP), consist of cell-free culture supernatant of avirulent BA strains precipitated using aluminium or adsorbed onto alhydrogel [38]. However, an intensive dosing regimen, lack of consistent production technique, and associated reactogenicity with these vaccines have led to the search for newer prophylactic candidates [37,39].

In this study, we have prepared a nanoformulation by the co-administration of aluminium hydroxide nanoparticles (AH nps) and D4- encapsulating non-ionic surfactant vesicle (NISV). The nanoformulation was evaluated for both antigen- specific humoral and cellular response immune response as well as its protective efficacy against BA spore challenge in a mouse model.

## 2. Materials and Methods

### 2.1. Materials

All buffers were prepared in sterile deionised water. Ni-NTA was purchased from Qiagen, Hilden, Germany. Luria-Bertani (LB) was purchased from Difco, MA, USA Ammonium chloride, Span 60, cholesterol, LPS, sodium bicarbonate, sodium phosphate monobasic, and sodium phosphate dibasic was purchased from Sigma Aldrich, St. Louis, MO, USA. L-glutamine, urea, and sodium chloride (NaCl) were purchased from Amresco, OH, USA. Isopropyl β-D-thiogalactopyranoside (IPTG) was purchased from Hi-media. HEPES, streptomycin and ampicillin were purchased from USB, OH, USA. Chloroform was purchased from Fischer scientific, MA, USA. FITC was purchased from MP Biomedicals, CA, USA. Bradford reagent was purchased from Biorad, CA, USA. The tissue culture plate and micro BCA kit were purchased from Thermo Fisher Scientific, Waltham, Massachusetts MA, USA. FBS was purchased from Gibco, MA, USA. HRP-conjugated anti mouse IgG, IgG1, and IgG2a were purchased from Santa Cruz Biotechnology, TX, USA. TMB substrate was purchased from BD Biosciences Pharmingen, NJ, USA.

### 2.2. Purification of D4

Protective antigen domain 4 (D4) was expressed and purified as described by Manish et al. [25]. Briefly, D4 expression plasmid transformed *E. coli* cells were grown to an OD _600 nm_ of 0.8 at 37 ℃ before inducing with 1 mM isopropyl β-D-thiogalactopyranoside (IPTG) and allowed to grow in an incubator shaker for 6 h. Then, the cells were harvested at 5000× *g* for 10 min. The bacterial cell pellet was lysed and solubilised using denaturing lysis buffer (8 M urea, 0.1 M phosphate buffer pH 7.4, and 250 mM NaCl) on a rotary shaker for 2 h at RT. The insoluble fraction of cell lysate was removed by centrifugation at 13,000× *g* for 30 min at RT. The supernatant was incubated with Ni-NTA slurry for 2 h. This mix was transferred to a propylene tube column, and slurry bound D4 was renatured by passing a gradient of urea solution 8 M to 0 M (0.1 M phosphate buffer pH 7.4, 250 mM NaCl). The steps after 4 M urea gradient were carried out at 4 ℃. The column was washed with 10 bed volume of 10 mM, 20 mM, and 30 mM imidazole, 250 mM NaCl containing 0.1 M phosphate buffer pH 7.4. The column bound protein was eluted using 300 mM Imidazole containing 0.1 M phosphate buffer pH 7.4 and 250 mM NaCl. Protein content was quantified using Bradford reagent (Biorad, CA, USA).

### 2.3. Preparation of Vaccine Formulation

#### 2.3.1. Preparation of D4-Encapsulated NISV (NISV-D4)

D4-encapsulated NISV was prepared and the protein content was determined as described in our previous publication [40]. Briefly, 5.5 mg of Span 60 and 7.4 mg of cholesterol were dissolved in 5 mL of chloroform in a round-bottom flask. Subsequently, 1 mL of D4 (2 mg/mL) in 0.1 M phosphate buffer pH 7.4 was added to the flask and vortexed for 60 s. Then, the mixture was emulsified by sonicating in a water bath sonicator for 5 min with on/off pulse. The organic phase was evaporated in a rotary evaporator at 60 rpm at RT. The resulting thin film was resuspended in PBS. Unentrapped protein was separated by ultracentrifugation. Protein encapsulation was determined by micro-BCA. Unencapsulated protein was removed by ultracentrifugation, and the obtained pellet fraction was treated with 1% sodium dodecyl sulfate (SDS) and centrifuged at 13,000× *g*. The encapsulated protein was quantified by a micro-BCA kit as per the manufacturer’s protocol. Protein concentration was quantified from the standard curve of BSA in 1% SDS.

#### 2.3.2. Preparation of Aluminium Hydroxide Nanoparticles (AH nps)

In our previous publication [41], we have described the preparation and quantification of aluminium hydroxide nanoparticles. Briefly, equal volumes of 0.06 M AlCl_3_ were mixed with 0.18 M NaOH with continuous stirring in a magnetic stirrer. The cloudy suspension was centrifuged, washed with sterile deionised water, and finally resuspended in 0.1 M phosphate buffer pH 7.4.

#### 2.3.3. Preparation of Combinatorial NISV-D4/AH np/alhydrogel Vaccine Formulation

NISV-D4 encapsulating 250 μg/mL D4 was centrifuged and resuspended in AH np suspension/alhydrogel consisting of 250 μg/mL of aluminium. Each mouse was immunised with a formulation consisting of NISV encapsulating 25 μg D4 and adsorbed with AH np (NISV-D4/AH np) or with alhydrogel (NISV-D4/alhydrogel).

#### 2.3.4. In Vitro Release Assay

In order to obtain the release kinetics of D4, adjuvant formulations were mixed with 1 × PBS (phosphate buffer saline) in a 1:3 volume of vaccine formulation and PBS. The formulation PBS samples were kept at 37 ℃ with gentle shaking. One mL of the samples was obtained at different time intervals and supernatant was collected after centrifugation. The pellet obtained after centrifugation was resuspended in PBS and mixed with the stock sample. The D4 content in the supernatant was determined by micro-BCA as per the manufacturer’s protocol.

### 2.4. Immunological Assays

#### 2.4.1. Mice Immunisation

Four to 6-week-old Swiss albino mice were procured from the Central laboratory for animal resources, Jawaharlal Nehru University, New Delhi. Each group consisted of 3 mice each and were grouped as PBS, D4 only, AH np/D4, alhydrogel/D4, NISV-D4, AH np/NISV-D4, and alhydrogel/NISV-D4. Mice groups were immunised via intraperitoneal route (i.p.) on day 0, 14, and 28 with 100 μL of the formulation consisting of 25 μg D4. Mice were bled on day 14, 28, and 42 post immunization via the retro-orbital route. Serum was separated by centrifugation at 10,000× *g* at 4 °C and stored at −20 °C for further use.

#### 2.4.2. Determination of Anti-D4 Antibody and Its Isotypes

Anti-D4 antibody titers (total IgG) and isotypes (IgG1 and IgG2a) was determined by ELISA in the sera of the immunised mice. Briefly, 500 ng D4 was coated into a 96-well micro-titer plate and incubated overnight at 4 ℃ in a humidified atmosphere. Then, the plates were blocked with 2% BSA in 37 ℃ for 1 h followed by three washings with PBST (0.05% tween 20). This was followed by the addition of 100 μL of two-fold serial dilution of serum samples. The plate was incubated at 37 ℃ for 2 h, followed by three washings with PBST. The captured antibody was detected using a secondary antibody, anti-mouse IgG HRP conjugate (Santa Cruz), for 1 h at 37 ℃. Enzymatic activity was estimated using TMB as the substrate. The reaction was stopped using 1N HCl, and absorbance was noted at 450 nm in an ELISA plate reader (TECAN). The endpoint titer was calculated as the reciprocal of the highest dilution having absorbance above the cut-off. The cut-off is the mean OD value plus three times the standard deviation of the PBS control group in each respective dilution. Any sample dilution having absorbance higher than the cut-off value was considered a positive reading.

Similarly, IgG1 and IgG2a antibody titers were determined by using secondary anti-mouse IgG1 or IgG2a HRP-conjugated antibody [40].

#### 2.4.3. Isolation of Splenocytes

On the 14^th^ day post immunisation schedule, mice were sacrificed by cervical dislocation, and spleens were aseptically removed and dissociated by grinding under the frosted ends of two microscopic slides to prepare a single cell suspension. Erythrocytes were lysed with 0.9% ammonium chloride and splenocytes isolated via centrifugation. Splenocytes were washed thrice with incomplete RPMI and viability determined by the trypan blue exclusion method. Splenocytes were suspended in complete RPMI media supplemented with 10% FBS.

#### 2.4.4. T-cell Restimulation

Splenocytes were isolated as described above and were plated in a 96-well micto-titer plate at a concentration of 1 × 10^5^ cells/well. The cells were stimulated with 5 μg of Concavalin A (positive control), media only (negative control), or 5 μg D4 (test sample) and incubated at 37 ℃ with 5% CO_2_. Splenocyte supernatants were harvested post 48 h of stimulation, and cytokine levels were determined by ELISA as per the manufacturer’s protocol.

#### 2.4.5. Evaluation of Ex Vivo Cytokine Levels

Antigen-specific cytokines IL-2, IL-4, IL-6, IL-10, IFN-γ, and TNF-α were determined using a BD OptEIA kit according to the manufacturer’s protocol. Briefly, a 96-well micro-titer plate was coated with the capture antibody of the respective cytokine in compatible buffers and incubated overnight at 4 ℃. The plate was aspirated, washed thrice with 1x PBST, and blocked with 10% FBS in 1x PBS for 1 h at room temperature (RT). After the incubation, the solution was aspirated and washed with PBST. Post washing, the plate was incubated with the harvested splenocyte supernatant for 2 h at RT. Then, the plate was aspirated and washed with PBST thrice. The plate was incubated with detector (anti-mouse IgG-HRP) for 1 h at RT. Following this, the plate was aspirated and washed with PBST seven times and incubated with TMB substrate in dark. The reaction was stopped by adding 50 μL of 1 N HCl, and absorbance was noted at 450 nm in an ELISA plate reader. Cytokine concentrations were determined using a linear regression equation obtained from the absorbance values of the standard curve.

#### 2.4.6. Anthrax Spore Challenge

Each mice group (*n* = 10) was immunised thrice with vaccines containing PBS or D4 only or AH np/D4 or alhydrogel/D4 or NISV-D4, AH np/NISV-D4 or alhydrogel/NISV-D4 in two-week intervals between immunisation. Two weeks after the last immunisation, the mice were challenged with 0.5 × 10^3^ spores of a clinical virulent strain of *Bacillus anthracis* via the intraperitoneal (i.p.) route. The challenged mice were kept in an isolator in a BSL3 facility and observed for 14 days for death and morbidity. Percentage survival of the vaccine formulations was compared and represented by plotting a Kaplan–Meir curve.

### 2.5. Statistical Analysis

GraphPad Prism v6.05 software was used for data preparation and statistical analysis. The results are represented as the mean with the standard error of mean (SEM) from each treatment group. For statistical analysis between vaccinated mice groups, either one-way or two-way ANOVA followed by Tukey’s multiple comparison tests were employed. *p* < 0.05 was considered as significant. * <0.05, **<0.01, ***<0.001, ****<0.0001. ns represent non-significant.

## 3. Results

### 3.1. In Vitro Release Kinetics

In vitro release assay provides an insight into the probable pharmacokinetics of a drug/antigen release from a nanoparticle. Samples were collected at different time periods and analysed for D4 content by micro-BCA assay. Figure 1 shows the D4 release profile from the different adjuvant formulations with time. After the initial burst release of 40% at day 1, NSIV-D4 showed cumulative release of 80% at day 6. Meanwhile, AHnp-D4 showed 10% of release at day 1, which followed approximately 15% of cumulative release at day 6. A similar fashion of release profile was seen with commercial alhydrogel+D4, which showed 10% of release at day 1 followed by a cumulative release of 15% at day 6. Admixing NSIV-D4 with AHnp significantly reduced the initial burst release to 20% at day 1 and almost 60% release at day 6, while alhydrogel/NISV-D4 exhibited a similar 60% cumulative release at day 6. The addition of AHnp with NSIV-D4 affected the initial burst from NSIV-D4 alone by 50% at day 1.

### 3.2. Adjuvant Effect of the Nanoformulation for Induction of Humoral Response Against D4

As a function to measure humoral response, the induction of D4-specific antibodies across all the formulations was determined by indirect ELISA. Mice groups (*n* = 3) were immunised with D4 only, NISV-D4, AHnp/D4, alhydrogel/D4, AHnp/NISV-D4, or alhydrogel/NISV-D4. Prime immunisation was followed by two booster doses on day 14 and 28. Antibody titer was determined from mice serum by ELISA. D4 only did not generate any detectable antibody titer even after three immunisations. When immunised with a single adjuvant by either encapsulation in NISV or adsorption onto AH np or alhydrogel, the highest antibody titer was produced by NISV-D4 when compared across the different time periods among the single adjuvant immunised mice groups. There was a gradual increase in the antibody production by NISV-D4 across day 14, 28, and 42 as can be seen from Figure 2. NISV-D4 (mean antibody titer value: 1, 61,000 ± 34) enhanced 2-fold anti-D4 antibody titer production as compared to AHnp/D4 (mean antibody titer value: 76,000 ± 18) (*p* < 0.0001) and 10-fold anti-D4 antibody titer when compared to alhydrogel/D4 (mean antibody titer value: 15,000 ± 24) (*p* < 0.0001) on day 42. However, immunisation of AH np/NISV-D4 enhanced the antibody titer (mean antibody titer value: 3, 20,000 ± 53) induction by almost 2-fold as compared to NISV-D4 (mean antibody titer value: 1, 61,000 ± 34) (*p* < 0.0001). However, alhydrogel/NISV-D4 (mean antibody titer value: 71,000 ± 34) produced antibody titers comparable to AHnp/D4 (mean antibody titer value: 76,000 ± 18) (*p*: ns), but it did not enhance the antibody production as compared to its counterpart NISV-D4/AH np (*p* < 0.0001). However, alhydrogel/NISV-D4 (mean antibody titer value: 71,000 ± 34) did increase the D4-specific antibody titer as compared to alhydrogel/D4 (mean antibody titer value: 15,000 ± 24) (*p* < 0.0001).

### 3.3. IgG Isotypes to D4 in Response to Co-Administration of AH np/NISV-D4

Antibody isotypes play a major role in activating host complement components and facilitate the clearance of pathogen from the body. We evaluated D4-specific IgG isotypes, IgG1 and IgG2a respectively, in response to NISV-D4/AH np and other control groups. D4 alone was unable to detect any significant antibody isotype, even after three immunisations. When administered with alhydrogel, D4 primarily elicited IgG1 antibody as reported in previous studies using D4 as antigen [41]. However, AH np enhanced the IgG2a production against D4. This is in agreement with our previous results, where it was shown that AH np promotes IgG2a production against D4 [41]. Similarly, NISV-D4 promoted IgG2a production, which is in agreement with our previous results [40]. In this study, the novel combinatorial nanoformulation AH np/NISV-D4 yielded a mixed IgG1/IgG2a response. From Figure 3a,b, it is observed that combinatorial administration of the nanoformulation significantly boosts both the D4-specific IgG1 and IgG2a response along the course of detection as compared to the control groups. Interestingly, combinatorial administration of alhydrogel/NISV-D4 also significantly enhances the D4-specific IgG2a response as compared to only alhydrogel/D4 (*p* < 0.001), which primarily elicits an IgG1 response. This enhancement in the IgG2a can be attributed to NISV, which facilitates antigen uptake by professional antigen-presenting cells.

### 3.4. Combinatorial Nanoformulation NISV-D4+ AH np Stimulates a Th1/Th2 Cytokine Profile

Naïve immature antigen-presenting cells phagocytose antigens via different mechanisms and process the antigen to present its different epitopes to a naïve T cell which directs the naïve T cell to stimulate the Th response. On antigen uptake and maturation, antigen-presenting cells produce cytokines in the mileau, which directs a naïve T cell towards a particular Th phenotype. Cytokines such as IL-2 and IFN-γ promote Th1 immune response while the secretion of cytokines such as IL-4 directs the T cell towards a Th2-biased immune response. In order to study the T-cell response initiated by all the different vaccine formulations against D4 in the immunised mice, we checked *ex vivo* Th1/Th2 cytokine secretion in the splenocyte culture supernatant after re-stimulating the cells with D4. Th1 cytokines 4a–c (IL-2, IFN-γ, TNF-α) and Th2 cytokines 4d–f (IL-4, IL-6, IL-10) were determined by ELISA. As seen from Figure 4, splenocytes of mice immunised with alhydrogel/D4 immunised mice produced primarily Th2 cytokines, which can be seen from the elevated levels of IL-4, IL-10, and IL-6 (Figure 4d–f). As seen in Figure 4a–c, Th1 cytokine (IL-2, IFN-γ and TNF-α) production was not significantly induced in alhydrogel/D4 immunised mice. However, splenocytes from mice immunised with AH np/D4 were able to generate significant levels of both Th1 and Th2 cytokines as compared to alhydrogel/D4. This is consistent with our previous results, where we reported a mixed cytokine response by D4 in response to AH np/D4 immunisation [38]. When we looked into cytokine production in the splenocyte of NISV-D4 immunised mice, we found a mixed Th1/Th2 with Th1 cytokine-dominant response, as can be seen from the significant secretion of IL-2, TNF-α, and IFN-γ (Figure 4a–c). This result is also consistent with the antibody subtype response, where NISV-D4 elicited an IgG2a biased response. Moreover, all these results are in parallel with our previous results [37], where we showed the adjuvant potential of niosomes encapsulating PA and D4. When we evaluated the cytokine profile elicited by the co-administration of the AHnp/NISV-D4, we observed a substantial increase in the levels of Th1 cytokines IL-2, IFN-γ, and TNF-α (Figure 4a–c), when we compare with AH np/D4 (*p* < 0.00011, *p* < 0.01, *p*: ns), NISV-D4 (*p* < 0.0001, *p* <0.0001, *p* < 0.01), and alhydrogel/NISV-D4 (*p* < 0.0001, *p* < 0.01, *p* < 0.0001). However, among the Th2 cytokine profiles, NISV-D4/AH np did not enhance the production of IL-6, as could be seen in the case of alhydrogel/D4 (*p* < 0.0001) and alhydrogel/NISV-D4 (*p* < 0.0001), which produced significant levels of IL-6. Even IL-4 levels secreted by NISV-D4/AH np were comparable with all the control. Thus, it indicates that the combinatorial adjuvant system NISV/AH np has the potential to augment a primarily Th1-biased immune response without suppressing Th2 cytokine production.

### 3.5. Combinatorial Nanoformulation of NISV-D4/AH np Augments Superior Protection against Anthrax in Mice

Two weeks after the last immunisation, mice (*n* = 10) from each immunised group were challenged with the virulent strain of *Bacillus anthracis* and observed for death and morbidity to the next 14 days. As shown in Figure 5, control mice groups receiving PBS or D4 antigen alone succumbed to death due to infection within 2–3 days post infection. While the mice groups immunised with alhydrogel/D4 and AH np/D4 showed 30% and 40% protection, respectively, NSIV-D4 was able to offer 50% protection. By contrast, combining AH np with NSIV-D4 showed enhanced protection to 70% at the end of 14 days, whereas the alhydrogel/NSIV-D4 mice group exhibited 55% protection.

## 4. Discussion

With constant threat of emerging pathogens, the quest to develop new vaccine strategies has continuously evolved. With the advent of nanotechnology, nanoparticle-based vaccines have received a great amount of attention, due to their enhanced uptake efficacy by antigen-presenting cells, ease of engineering physicochemical properties, and targeted delivery [11,42,43]. To date, the most widely used vaccine adjuvant in attenuated or inactivated human vaccine is alum/aluminium salts, and it has been remarkably effective in generating neutralising antibodies against invading pathogens. The adjuvant activity of aluminium has never been conclusive and it has been reported to mediate its adjuvant potential through the release of host DNA [44,45], NLRP3 inflammasome [46,47], prostaglandin E2 production [48], uric acid production [49], or interacting with the cell lipid membrane [50]. Despite these indecisive modes of action of alum, it is well documented that aluminium primarily promotes humoral response and fails to elicit a cellular response, particularly CD4^+^ Th1 response and CD8^+^ CTL response [51,52]. Hence, new strategies to enhance and tailor the immune response by aluminium along with co-stimulatory adjuvants have been constantly carried out. One such licensed adjuvant system, AS04, comprises of a TLR4 agonist MPL combined with aluminium [7,53], which is being currently used in cervarix (Human papilloma virus vaccine) and fendrix (Hepatitis B vaccine). Apart from this, different combinatorial adjuvant systems such as liposome/alhydrogel [54] and chitosan nanoparticle/alhydrogel [55] are currently being studied as a new generation of adjuvants and have shown very promising results. In this study, we have developed a novel combinatorial nanoformulation by the co-administration of NISV and AH np and evaluated its efficacy against *B. anthracis* antigen D4. Span 60 and cholesterol have been reported to be used for the preparation of NISV with very high antigen entrapment while maintaining the antigen integrity as reported in our previous publication [40]. The antigen release profile provides relevant information regarding the probable pharmacokinetics of the formulation *in vivo.* D4-encapsulating NISV showed a burst release in the initial 24 h of analysis releasing around 60% of the antigen, followed by a gradual release of the antigen up to 80% by the 6^th^ day of analysis. Alhydrogel had the slowest antigen release with around 15% antigen release by the 6^th^ day of analysis, followed by AH np with only around 15% antigen release. This can be correlated to the higher binding coefficient of D4 to alhydrogel and AH np, as was reported in our previous article and by other groups as well [41,56,57]. The D4 release kinetics from NISV-D4+ AH np and NISV-D4+ alhydrogel was almost similar. The initial burst release can be correlated with the antigen release from within NISV. However, upon elution, they bind to the particlulate alhydrogel/AH np, thereby slowing the antigen release to the buffer. Efficient antibody production by plasma B cells requires a vaccine antigen to be phagocytosed by professional antigen-presenting cells (APC) such as dendritic cells (DC) and macrophages (Mac) and present the processed peptide via MHC class II receptor to its cognate B cell or T cell receptor, leading to an enhanced and durable antibody production as well as a cellular response. Nanoparticles have been reported to be efficient antigen-presenting vectors to APCs, thereby generating a robust humoral and cellular response [10,58,59,60].

Alhydrogel potentially provides its surface area for binding to antigen or additional adjuvants. However, particles of alhydrogel that are in the micron range do not typically offer a large area for binding, as it has been reported that only 1.7% of the area of alhydrogel is available for binding [61]. In our previous publication, we had shown that we can improve the available surface area for binding to antigens by synthesising aluminiujm hydroxide nanoparticles [41]. Moreover, APCs are more efficient in the phagocytosis of nanoparticulate adjuvants as compared to micron-range adjuvants [10]. Protective antigen domain 4 (D4) is the most immunodominant antigen of protective antigen, but D4 alone is insufficient to generate a robust humoral and cellular response [28]. In our study, we evaluated the humoral response generated by NISV+AH np against D4 and showed that the combinatorial nanoformulation of NISV-D4/AH np facilitated a robust D4-specific antibody production as compared to its control NISV-D4/alhydrogel. We also looked into the IgG1/IgG2a isotype response in the serum of immunised mice. A generation of mixed IgG1/IgG2a response by NISV-D4/AH np suggests that this combinatorial adjuvant system can facilitate the production of both Th1 and Th2 cellular response. T-cell response is initiated during antigen capture, processing, and peptide presentation by MHC class II to the TCR of naïve CD4^+^ T cells. This interaction mediates APCs to upregulate CD80/CD86 to interact with the CD28 of naïve CD4^+^ T cells. Antigen capture promotes the APC to release cytokine in the milieu, which directs the naïve CD4^+^ T cell to differentiate towards a Th1/Th2 cytokine-producing subtype. A T-cell dependent adaptive immune response also helps in the formation of T follicular helper (Tfh) cell and germinal center, thereby producing durable and high-affinity antigen-specific antibody response [62]. Th1/Th2 cytokine profiling was performed by euthanising the immunised mice and re-stimulating them with D4. Alhydrogel is known to facilitate Th2 cytokine response, primarily producing IL-4 and IL-10 thereby suppressing the proliferation of antigen-specific Th1 cytokine-producing cells [51]. A similar trend was observed in our experiment as well (Figure 4). However, the restimulation of splenocytes of AH np/D4 immunised mice with D4 produced both antigen-specific Th1 and Th2 cytokines, thereby demonstrating its potential to be a superior adjuvant candidate as compared to alhydrogel. Similarly, NISV encapsulating D4 was also able to stimulate the production of both antigen-specific Th1 and Th2 cytokine production. On analysing the cytokine profile of the combinatorial adjuvants, it was observed that AH np/NISV-D4 was able to stimulate T cells to secrete both Th1 as well as Th2 cytokine. However, on comparison with its counterpart alhydrogel/NISV-D4, it was observed to elicit a mixed Th1/Th2 cytokine response as compared to primarily Th2 cytokines by alhydrogel, which can be attributed to the presence of NISV. Thus, both the antibody isotype and Th1/Th2 cytokine results observed in this experiment imply that NISV/AH np as a combinatorial adjuvant system can elicit a significant D4 specific humoral as well as cellular response. Although the exact mechanism by which NISV/AH np can stimulate the immune response is unknown, a general mechanism can be believed to be generated by pinocytosis of the particles by phagocytes, leading to the secretion of danger signals such as uric acid, hsp 70, or damaged DNA from necrotic cells, leading to a proliferation of cytokines, chemokines, inflammation-recruited neutrophils, and monocytes, leading to an enhanced antigen uptake and hence the elevated response.

In summary, we have developed an interesting nanoformulation that can potently program the Th as well as humoral response. This can be further implemented to enhance the efficacy of suboptimal vaccines or even improve the efficacy of pre-existing vaccines.

## 5. Conlusions

In this manuscript, we present an interesting nanoformulation comprising of protective antigen domain 4 (D4) encapsulated in non-ionic surfactant vesicles and co-administered with aluminium hydroxide nanoparticles. This nanoformulation significantly elevated the humoral antibody response of D4, as compared to the control alhydrogel immunised group. D4-specific cytokines suggested that the nanoformulation could potentially stimulate naïve CD4^+^ T cells to mature CD4^+^ T cells producing Th1 and Th2 cytokines. This nanoformulation also enhanced protective efficacy against the anthrax spore challenge as compared to control alhydrogel.

## Figures and Tables

**Figure 1 vaccines-08-00571-f001:**
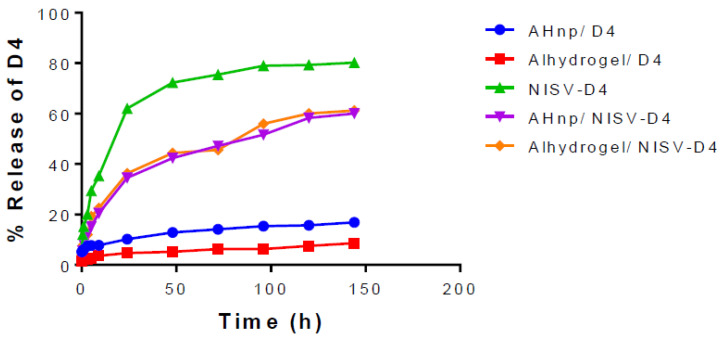
In vitro release kinetics. Nanoparticle formulations were suspended in 1 × PBS at 37 ℃. Samples were collected at different time intervals and protein content estimated by micro-BCA. A slow release of D4 can be observed from alhydrogel and aluminium hydroxide nanoparticles (AH np), while an initial burst release of D4 can be observed in the case of non-ionic surfactant based vesicles encapsulating D4 (NISV-D4). The co-administration of NISV and AH np/alhydrogel resulted in a sustained and gradual release of D4 into the mileau.

**Figure 2 vaccines-08-00571-f002:**
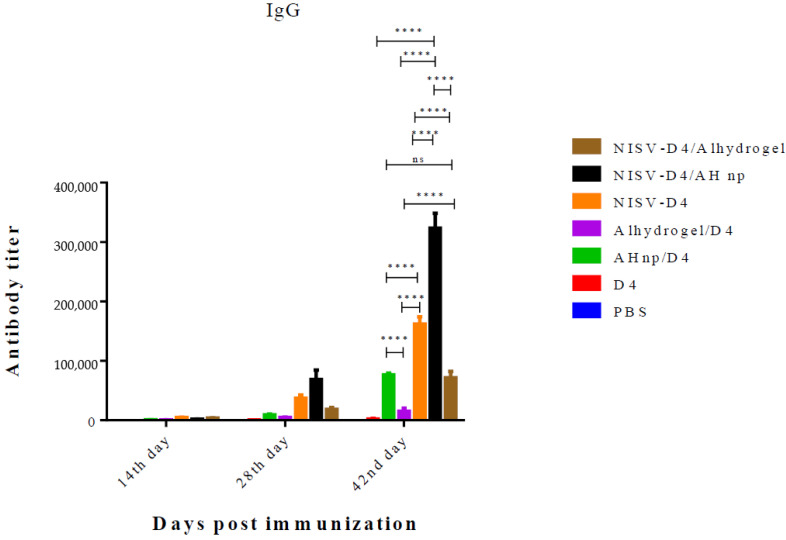
Anti-D4 IgG titres elicited by Swiss albino mice immunised via intraperitoneal (i.p.) route with PBS, D4 only, AH np/D4, alhydrogel/D4, NISV-D4, NISV-D4/alhydrogel, and NISVD4/AH np. Subsequent booster doses were administered on days 14 and 28. Sera were collected from individual mice and analysed in triplicate for D4-specific IgG antibodies. The results were expressed as mean values with standard deviation (SD) from each individual mice group. The *p*-value was calculated between AH np adjuvanted groups and alhydrogel using two-way Anova followed by Tukey’s multiple comparison tests. ns: non-significant, **** *p* < 0.0001.

**Figure 3 vaccines-08-00571-f003:**
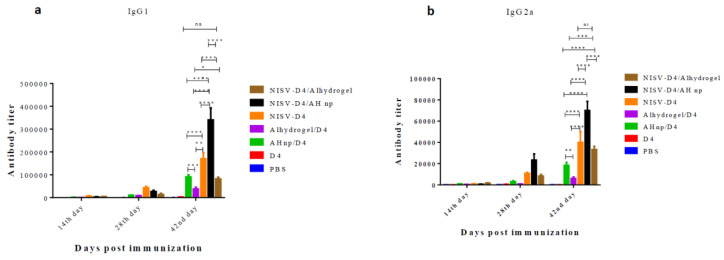
Anti-D4 (**a**) IgG1 and (**b**) IgG2a titres elicited by Swiss albino mice immunised via IP route with PBS, D4 only, AH np/D4, alhydrogel/D4, NISV-D4, NISV-D4/alhydrogel, and NISV-D4/AH np. Subsequent booster doses were administered on days 14 and 28. Sera were collected from individual mice and analysed in triplicate for D4-specific IgG antibodies. The results were expressed as mean value with standard deviation (SD) from each individual mice group. *p*-value was calculated between AH np adjuvanted groups and alhydrogel using two-way Anova followed by Tukey’s multiple comparison tests. * *p* < 0.05, ** *p* < 0.01, *** *p* < 0.001, **** *p* < 0.0001, ns represents non-significant.

**Figure 4 vaccines-08-00571-f004:**
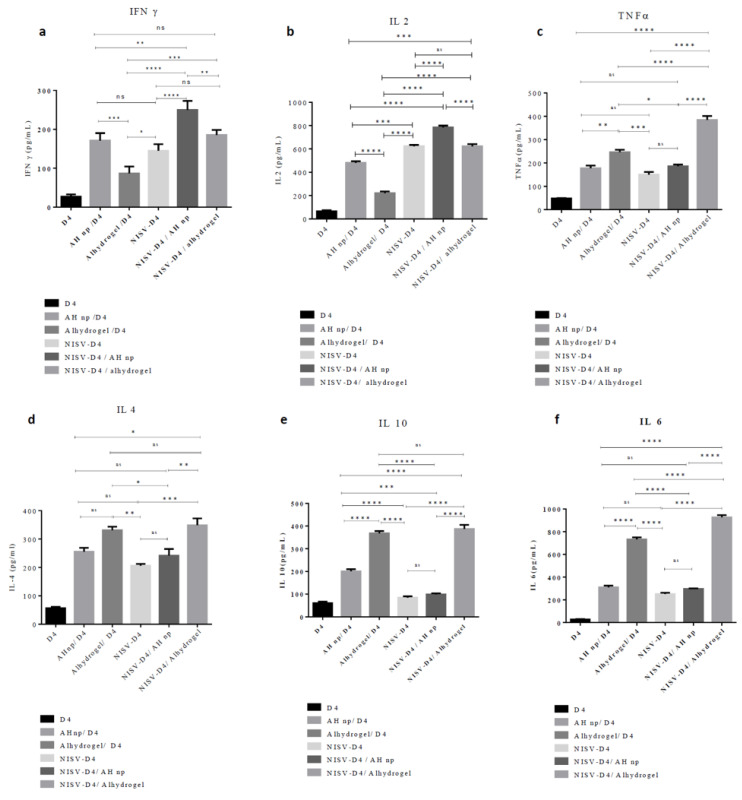
Immunisation with NISV-D4/AH np elicits a mixed Th1/Th2 immune response. Evaluation of cytokine (**a**) IFN-γ, (**b**) IL-2, (**c**) TNF-α, and (**d**) IL-4, (**e**) IL-10, and (**f**) IL-6 from splenocyte culture supernatant post immunisation and stimulated in vitro with D4 or media only for 48 h. Error bars represent ± SD of three experiments. Statistically significant change between immunised groups were calculated using two-way ANOVA followed by Tukey’s multiple comparisons test * *p* < 0.05, ** *p* < 0.01, *** *p* < 0.001, **** *p* < 0.0001, ns: non-significant.

**Figure 5 vaccines-08-00571-f005:**
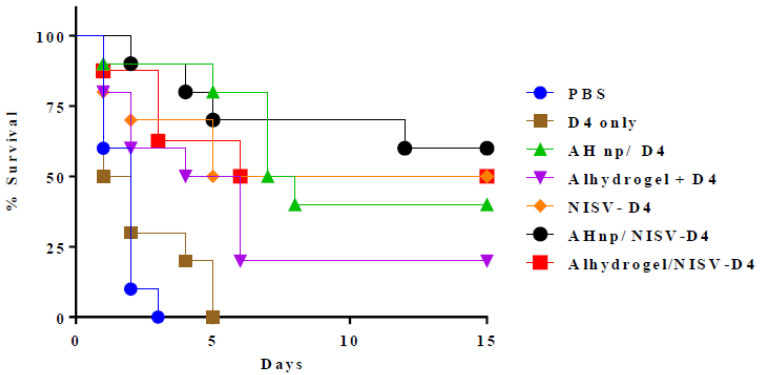
Protective efficacy of the vaccine formulations against BA spore challenge was checked in Swiss albino mice. Fourteen days after the 2nd booster, immunised mice were administered with 0.5 × 10^3^ spores of a virulent strain of BA. Infected mice were kept in an animal isolator in BSL 3 and observed for death and morbidity for 14 days.

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
