# Peer review of "Co-Administration of Aluminium Hydroxide Nanoparticles and Protective Antigen Domain 4 Encapsulated Non-Ionic Surfactant Vesicles Show Enhanced Immune Response and Superior Protection against Anthrax"

_vaccines, 2020, doi:10.3390/vaccines8040571_

Round 1
Reviewer 1 Report
This manuscript has illustrated the aluminium hydroxide nanoparticles (AH np)and non-ionic surfactant based vesicles (NISV) in up-regulation of the vaccine. The experimental results have completely clarified the induction of the immune reaction of this nanoformulation.
However, some for suggestions as following:
- The combination of aluminium hydroxide nanoparticles (AH np) and non-ionic surfactants has been reported in many studies. The novelty of this nanoformulation should be illustrated and emphasized in the Introduction section.
- 2. The cytotoxicity of this nanoformulation should be tested in comparison of aluminium hydroxide nanoparticles (AH np).
Author Response
- The combination of aluminium hydroxide nanoparticles (AH np) and non-ionic surfactants has been reported in many studies. The novelty of this nanoformulation should be illustrated and emphasized in the Introduction section.
Response: We agree with the reviewers suggestion that combinatorial formulations involving alhydrogel/ aluminium hydroxide gel has been reported, are under clinical trials and even licensed (AS-04). However, as per our knowledge, this combinatorial formulation involving aluminium hydroxide nanoparticle and non-ionic surfactant vesicle has never been reported by any group. To further lay down emphasis on the novelty of our combinatorial formulation, we have incorporated a paragraph in our introduction section. (Lines:76-79 )
- The cytotoxicity of this nanoformulation should be tested in comparison of aluminium hydroxide nanoparticles (AH np).
Response: We agree with the reviewer’s concern, but due to COVID pandemic, we are currently under lockdown, and carrying out this experiment won’t be feasible for us. However, in our earlier manuscripts, we have tested the cytotoxicity of the individual nanoparticles in RAW 264.7 macrophage cells, and they were well tolerated with more than 90% viability.
Reviewer 2 Report
Gogo, et al present an interesting manuscript here describing the creation for a novel adjuvant formulation using well known and tolerated aluminum hydroxide adjuvant formulation, resulting in enhanced immune response to the antigen of choice. The authors previously reported on their success in synthesizing aluminum hydroxide nanoparticles as well as the delivery of the antigen domain 4 (D4) of Bacillus anthracis. In this study, the authors set out to evaluate the synergistic effect of aluminum hydroxide nanoparticles (AH np) and non-ionic surfactant based vesicles (NISV) in modulating immune response against D4.
By itself, D4 only did not generate any detectable antibody titer even after three immunizations. Their data shows that when immunized with a single adjuvant by either encapsulation in NISV or adsorption onto AH np or alhydrogel, the highest antibody titer was produced by NISV-D4 when compared across the different time periods among the single adjuvant immunized mice group. The Ah np + NISV-D4 formulation was superior in its antibody production.
The authors have shown, convincingly, in the current study, the development of a novel combination nano-formulation by co-adsorption of NISV and AHnp and evaluated it’s efficacy against B.anthracis antigen D4. The formulation seems to be characterized by very high antigen entrapment while maintaining the an integrity of the antigen. The antigen release profile revealed a 40% immediate release of antigen in vitro, followed by a slow release over time, theoretically making the antigen available for APC processing, once phagocytized. This characteristic could be important for future pharmacokinetics considerations.
The authors demonstrate here that AHnp and NISV-D4 promote IgG2a production against D4, while the novel combinatorial nano-formulation; NISV-D4/AH np yielded a mixed IgG1/IgG2a response. This could potentially induce a Th1 immune response without suppressing Th2 cytokine production. However, since no flow cytometry assays were performed to identify cell proliferation and subset identification from spleen cells, the authors cannot be certain of this very likely outcome.
The authors also demonstrated that combining AHnp with NSIV-D4 increased protection of mice infected with Bacillus spores, increasing survival/protection to around 70% at the end of two weeks. This was almost 20% greater than the Alhydrogel+NSIV-D4 formulation.
Minor revisions
1. Throughout the manuscript, the authors should place a hyphen between the alpha numeric names of all cytokines. For example, instead of IL-2, the authors have IL 2.
2. Lines 167 – 168 : The authors should remove the “th”and “nd” from day 14, 28 and 42 respectively. This is the incorrect use.
3. Line 183: The authors write “Similarly, Ig1G1…” This should be IgG1
4. On page 6, lines 230 to 236, the authors referred repeatedly to “day 10” in talking about the data in figure 1. However, the data in figure 1 is presented in hours and go up to 200h. This is approximately 8 days. The authors should correct the narrative and figure to reflect the time period being referred to.
5. For figure 4 on page 9 of the manuscript, since you are using the same color scheme in each of the cytokine panels here, you just need one legend to represent each formulation instead of the six identical ones you currently have.
6. Lines 346 to 349: The authors need punctuation in this legend. No periods or commas until the end of the description.
7. It would be useful to carefully review this manuscript for errors in grammar, syntax and overall sentence construction since there were simple errors throughout. For example, the use of “Till date …” instead of To date in line 354 and “AS04, comprises of a TLR4 …” in line 363 instead of writing “ASO4 is comprised of….”. or line 400 --- removing “ T cell/ cellular response is” since it distorts the meaning of the sentence. Correction of these errors will greatly enhance the reading flow of the manuscript.
8. It would have been nice to see T cell proliferation assays and identification of Th1/Th2 subsets via Flow Cytometry to augment the reported cytokine data. This would add strength to the manuscript and lead to a more definitive answer where the mechanism of action is concerned.
9. It would also be nice to see a diagram showing the model/mechanism of proposed activity of AHnp with NSIV-D4 compared to Alhydrogel+NSIV-D4 formulation from the authors. This would help the reader to more easily follow the formulation created and the proposed mechanism of action.
Author Response
- Throughout the manuscript, the authors should place a hyphen between the alpha numeric names of all cytokines. For example, instead of IL-2, the authors have IL 2.
Response: Hyphen has been added between alpha numeric names of cytokines.
- Lines 167 – 168 : The authors should remove the “th”and “nd” from day 14, 28 and 42 respectively. This is the incorrect use.
Response: th and nd has been removed from day 14, 28 and 42 on lines 173-174.
- Line 183: The authors write “Similarly, Ig1G1…” This should be IgG1
Response: Ig1G1 has been replaced with IgG1 in line 189.
- On page 6, lines 230 to 236, the authors referred repeatedly to “day 10” in talking about the data in figure 1. However, the data in figure 1 is presented in hours and go up to 200h. This is approximately 8 days. The authors should correct the narrative and figure to reflect the time period being referred to.
Response: The narrative has been corrected as mentioned by the reviewer in lines 236-243. The experiment was performed till 150 h, which corresponds to approximately 6 days.
- For figure 4 on page 9 of the manuscript, since you are using the same color scheme in each of the cytokine panels here, you just need one legend to represent each formulation instead of the six identical ones you currently have.
Response: We do appreciate this point. However, for the convenience of the readers, we preferred to go with this format.
- Lines 346 to 349: The authors need punctuation in this legend. No periods or commas until the end of the description.
Response: Punctuation has been added to lines 346-349, which has moved to lines 354-357.
- It would be useful to carefully review this manuscript for errors in grammar, syntax and overall sentence construction since there were simple errors throughout. For example, the use of “Till date …” instead of To date in line 354 and “AS04, comprises of a TLR4 …” in line 363 instead of writing “ASO4 is comprised of….”. or line 400 --- removing “ T cell/ cellular response is” since it distorts the meaning of the sentence. Correction of these errors will greatly enhance the reading flow of the manuscript.
Response: The following changes have been incorporated. (Lines: 362; 371; 408)
- It would have been nice to see T cell proliferation assays and identification of Th1/Th2 subsets via Flow Cytometry to augment the reported cytokine data. This would add strength to the manuscript and lead to a more definitive answer where the mechanism of action is concerned.
Response: We do agree with the reviewer’s concern. However, we did not have the required expertise and reagents during the course of the experiment. We will definitely look at it in the future.
- It would also be nice to see a diagram showing the model/mechanism of proposed activity of AHnp with NSIV-D4 compared to Alhydrogel+NSIV-D4 formulation from the authors. This would help the reader to more easily follow the formulation created and the proposed mechanism of action.
Response: Although we have not yet deciphered the exact differences in mechanism of AH np with NISV-D4 and alhydrogel with NISV-D4, we hypothesize that nanoparticle being readily phagocytosed by APC’s like dendritic cell promote APC maturation, and upregulation of MHC II receptors, and secreting cytokines like IL-4 and IL- 12 thereby promoting both Th1 and Th2 cytokines. We have proposed a mechanism and attached a schematics as supplementary figure 1.

Reviewer 3 Report
In this work, Bhatnagar and co-workers report a novel nanoformulation as a vaccine against anthrax. The formulation includes protective antigen domain 4 (D4) encapsulated inside span-60 and cholesterol and then mixed with aluminium hydroxide nanoparticles as an adjuvant. Interestingly, the authors demonstrated that this novel formulation induced higher antibody production and therefore superior protection against anthrax. Overall, the work is novel, interesting and add significant understanding and advance to the field of nanoparticle vaccines. Therefore, I strongly recommend publishing this manuscript on Vaccines after very few minor revisions.
- It remains unclear that the NISV- D4 absorbed on the aluminium hydroxide nanoparticles since I could not find data proving this point. As such, I suggest two options: (i) the authors may provide more characterizations (Cryo-TEM, DLS) showing the adsorption of NISV- D4 on aluminium hydroxide nanoparticles or (ii) the authors may change the title and the text to “co-administration”, which does not change the meaning of the work but still show the correct nature of the formulation.
- The sentence “With a progress in nanoscience, nanotechnology has benefited tremendously in the field of drug delivery and vaccine delivery” need a reference: doi.org/10.1002/smll.201801702
- The author may cite a relevant work (doi.org/10.1002/smll.202002861) for the sentence “Nanoscale materials like liposome, emulsions, virus like particles, 62 ISCOM’s, polymeric particles have received attention as potential delivery vehicles as well as 63 immunomodulators.”
- The authors may add a scheme showing the concept of the new nanoformulation.
- The particle size and charge of aluminium hydroxide nanoparticles NISV- D4 may be provided in the experimental so the readers don’t need to find the data in previous work.
- Could the authors please discuss a bit more about the release mechanism of D4 in different formulations showed in Figure 4? What trigged the release?
- Line 388, chare “are” to “area”
- The authors may elaborate more on the retro-orbital injection route chosen in this work. How does it compare to intramuscular injection typical used in vaccines?
Author Response
- It remains unclear that the NISV- D4 absorbed on the aluminium hydroxide nanoparticles since I could not find data proving this point. As such, I suggest two options: (i) the authors may provide more characterizations (Cryo-TEM, DLS) showing the adsorption of NISV- D4 on aluminium hydroxide nanoparticles or (ii) the authors may change the title and the text to “co-administration”, which does not change the meaning of the work but still show the correct nature of the formulation.
Response: We acknowledge the reviewer’s concern and have modified the title and text accordingly. We have replaced “co-adsorption” with “co-administration”. Due to COVID pandemic, we are under lockdown and it won’t be feasible for us to perform the characterization experiments. However, we will definitely have a look at this suggestion.
- The sentence “With a progress in nanoscience, nanotechnology has benefited tremendously in the field of drug delivery and vaccine delivery” need a reference: doi.org/10.1002/smll.201801702
Response: We have incorporated the above reference in line 63 (Reference # 61).
- The author may cite a relevant work (doi.org/10.1002/smll.202002861) for the sentence “Nanoscale materials like liposome, emulsions, virus like particles, 62 ISCOM’s, polymeric particles have received attention as potential delivery vehicles as well as 63 immunomodulators.”
Response: We have incorporated the above reference in line 65 (Reference# 62).
- The authors may add a scheme showing the concept of the new nanoformulation.
Response: We have incorporated a schematics describing the preparation of NISV-D4/ AH np nanoformulation and have attached it as supplementary figure 2.